

# GEMS ozone profile retrieval: impact and validation of version 3.0 improvements

Juseon Bak[1], Arno Keppens[2], Daesung Choi[3], Sungjae Hong[3], Jae-Hwan Kim[3], Cheol-Hee Kim[3], Hyo-Jung Lee[3], Wonbae Jeon[3], Jhoon Kim[4], Ja-Ho Koo[4], Joowan Kim[5], Kanghyun Baek[6], Kai Yang[6], Xiong Liu[7], Gonzalo G. Abad[7], Klaus-Peter Heue[8], Jean-Christopher Lambert[2], Yeonjin Jung[9], Hyunkee Hong[10], Won-Jin Lee[10]

[1]Institute of Environmental Studies, Pusan National University, Busan 46241, Republic of Korea
[2]Royal Belgian Institute for Space Aeronomy (BIRA-IASB), Brussels, Belgium
[3]Department of Atmospheric Sciences, Pusan National University, Busan 46241, South Korea
[4]Department of Atmospheric Sciences, Yonsei University, Seoul, Republic of Korea
[5]Department of Atmospheric Sciences, Kongju National University, Kongju, Republic of Korea
[6]Department of Atmospheric and Oceanic Science, University of Maryland, College Park, MD 20742, USA
[7]Smithsonian Astrophysical Observatory (SAO), Center for Astrophysics | Harvard & Smithsonian, Cambridge, MA 02138, USA
[8]Institut für Methodik der Fernerkundung am Deutschen Zentrum für Luft- und Raumfahrt (DLR), Oberpfaffenhofen, Germany
[9] Major of Spatial Information Engineering, Division of Earth and Environmental System Sciences, Pukyong National University, Busan, Republic of Korea
[10]National Institute of Environmental Research, Incheon 22689, Republic of Korea

*Correspondence to*: Juseon Bak (juseonbak@pusan.ac.kr), Jaehwan Kim(jaekim@pusan.ac.kr)

**Abstract.** This study presents the first comprehensive description of the operational GEMS (Geostationary Environment Monitoring Spectrometer) ozone profile retrieval algorithm and evaluates the performance of the recently reprocessed version 3.0 dataset. The retrieval operates in the 310–330 nm spectral range and yields total degrees of freedom for ozone ranging from 1.5 to 3. Although the vertical sensitivity is limited, GEMS achieves an effective vertical resolution of 5–10 km and is capable of separating tropospheric and stratospheric ozone layers. This work primarily highlights the substantial algorithmic and calibration enhancements introduced in version 3.0 over the previous version, including improvements to the slit function, wavelength calibration, and radiometric calibration. In particular, the irradiance offset has been a major issue affecting the accuracy of ozone profile and other Level 2 products. To address this, the measured irradiance is scaled relative to a high-resolution solar reference spectrum using a correction factor. Residual wavelength-dependent biases in the normalized radiance are further addressed through soft calibration. As a result, version 3.0 significantly reduces spectral fitting residuals, lowering them from 0.8% in version 2.0 to 0.2% under nominal conditions. This improvement also mitigates the altitude-dependent oscillating biases observed in the previous version, which included up to 40 DU overestimation in the troposphere and 20 DU underestimation in the stratosphere, when compared with ozonesonde observations. The version 3 ozone profiles show agreement within 10 DU of ozonesonde profiles, with a mean bias of −7.7% in tropospheric ozone columns and within 5% in the stratosphere. Furthermore, the retrievals capture day-to-day vertical ozone variability, as demonstrated by comparisons





with daily ozonesonde launches in February and March 2024. Integrated ozone columns derived from the profiles also show
improved consistency with ground-based total ozone measurements, yielding a mean bias of −3.6 DU and outperforming the
GEMS operational total column ozone product.

## 1 Introduction


Atmospheric ozone is a powerful greenhouse gas and air pollutant, harming human health and ecosystems in the
troposphere (Van Dingenen et al., 2009; Isaksen et al., 2009). In the stratosphere, ozone is essential for protecting life on Earth
by absorbing harmful ultraviolet (UV) radiation from the Sun (Solomon, 1999). It also plays a key role in maintaining the
Earth's radiative balance and stratospheric temperature structure (Monks et al., 2015). Monitoring both layers is vital for
understanding pollutant transport, regulating air quality, addressing climate change, and protecting environmental health.
The Geostationary Environmental Monitoring Spectrometer (GEMS) onboard the Korean GEO-
KOMPSAT(Geostationary Korea Multi-Purpose Satellite)-2B satellite provides high temporal and spatial resolution data on
ozone, its precursors ($NO_2$ and $HCHO$), $SO_2$, and aerosols over East Asia (Kim et al., 2020). GEMS offers two primary ozone
products: total column ozone ($O_3T$) and the full ozone profile ($O_3P$). The $O_3T$ product is retrieved using the historical TOMS
look-up table algorithm (Kim et al., 2024), while the $O_3P$ product provides vertically resolved ozone information across 24
atmospheric layers, retrieved based on an optimal estimation-based inversion framework (Bak et al., 2020). Baek et al. (2023,
2024) provided a comprehensive evaluation of the GEMS v2.0 $O_3T$ product, examining its spatial and temporal
representativeness on hourly, daily, and seasonal scales through cross-comparisons with ground-based Pandora measurements
and independent satellite observations from polar-orbiting platforms. The product exhibited strong correlations with Pandora
(0.97) and satellite data (0.99), but showed a pronounced seasonal and latitudinal dependence in mean bias, which was
attributed to the absence of a calibration component accounting for the bidirectional transmittance distribution function (BTDF)
in irradiance measurements (Kang et al., 2024). A minor update to the look-up table was subsequently implemented, resulting
in the release of version 2.1 (Kim et al., 2024). Although the GEMS $O_3P$ product has not yet been fully described in peer-
reviewed literature, the algorithm implemented for processing version 2.0 closely follows the Smithsonian Astrophysical
Observatory (SAO) ozone profile algorithm used for generating the Ozone Monitoring Instrument (OMI) Collection 3 ozone
profile research product (Liu et al., 2010). The OMI ozone profile product has demonstrated its reliability in supporting studies
of ozone variability driven by the chemical and dynamical processes, quantifying global tropospheric budget of ozone, and
evaluating model representation. However, the project *Product Evaluation of GEMS L2 via Assessment with S5P and Other*
*Sensors (PEGASOS, funded by the European Space Agency)* reported the need for improvements prior to scientific use, citing
significant altitude-dependent oscillating biases in the GEMS $O_3P$ version 2.0 product, with deviations of up to 30 % in the
troposphere and from -10 % to -20% in the stratosphere (Keppens et al., 2024). In addition, the PEGASOS report identified
large discrepancies between the GEMS $O_3P$ and $O_3T$ products. The inconsistencies in ozone profile quality between GEMS





and OMI can be attributed to differences in radiometric and wavelength calibration stability, rather than to the retrieval
algorithm itself, which shares similar forward and inverse processes.
These findings motivated the development of version 3.0 of the GEMS ozone profile product, which incorporates
improvements in spectral and radiometric calibration, including:
(1) updating of the pre-flight measurements of slit functions to on-orbit derivations.
(2) correction of wavelength shifts in both radiance and irradiance spectra,
(3) implementation of irradiance offset correction to address solar diffuser–induced seasonal variation and long-term
optical degradation, and
(4) application of soft calibration to correct residual radiometric biases in the normalized radiances.

In addition to these calibration enhancements, the algorithmic updates include modifications to the forward model
calculations, fitting parameters, and several auxiliary inputs. This paper is structured around three main objectives. The
retrieval algorithm and the updates from version 2.0 to version 3.0 are introduced in the second section. Section 3 focuses on
retrieval characterization and error analysis based on optimal estimation diagnostics. Validation results using independent
reference datasets are discussed in Section 4. The final section concludes this paper, with remarks for future updates.

## 2. GEMS Ozone Profile Retrieval Algorithm
### 2.1 GEMS operations
GEMS is an ultraviolet-visible imaging spectrograph equipped with a single two-dimensional charge-coupled device (CCD)
array detector, with one dimension for 1,033 wavelengths and the other for 2,048 spatial pixels (Lee et al., 2024). It measures
solar irradiance once each night and Earth's backscattered radiance hourly from 07:45 to 16:45 Korea Standard Time (KST),
covering the spectral range from 300 to 500 nm with a spectral resolution of approximately 0.6 nm full width at half maximum
(FWHM). A shared optical path is used for both radiance and irradiance measurements, except for dedicated solar diffusers,
which operate on different duty cycles (daily and monthly) to manage sunlight intensity and prevent detector saturation. In
GEMS, the spatial pixels represent fixed ground-based observation points on Earth, aligned along the north–south direction
from a geostationary orbit and covering latitudes from 5° to 45°S, while in polar-orbiting satellites, the term "cross-track pixel"
is typically used, reflecting their spatial alignment across the flight path. In Earth observation mode, GEMS scans an east–
west swath from 75° to 145°E in approximately 700 mirror steps (scan lines) during full-scan mode and 350 mirror steps
during half-scan mode. Four scan modes — Half East (HE), Half Korea (HK), Full Central (FC), and Full West (FW) — are
operated sequentially, with their spatial extents shown in Figure 1 and the detailed operation schedule summarized in
Supplementary Tables 1 and 2. The operational data record begins on November 2, 2020, marking the start of the official




observation period. Currently, Version 2 Level 1C irradiance and radiance products are commonly used as inputs for Level 2
processing. To enhance computational efficiency and improve the signal-to-noise ratio, Level 1C and selected Level 2 products
(e.g., cloud, surface reflectance, total ozone) are also provided with spatial binning of 2×2 or 4×4 pixels. The ozone profile
retrieval specifically utilizes 4×4 binned data, resulting in a 512 × 175 frame dataset.

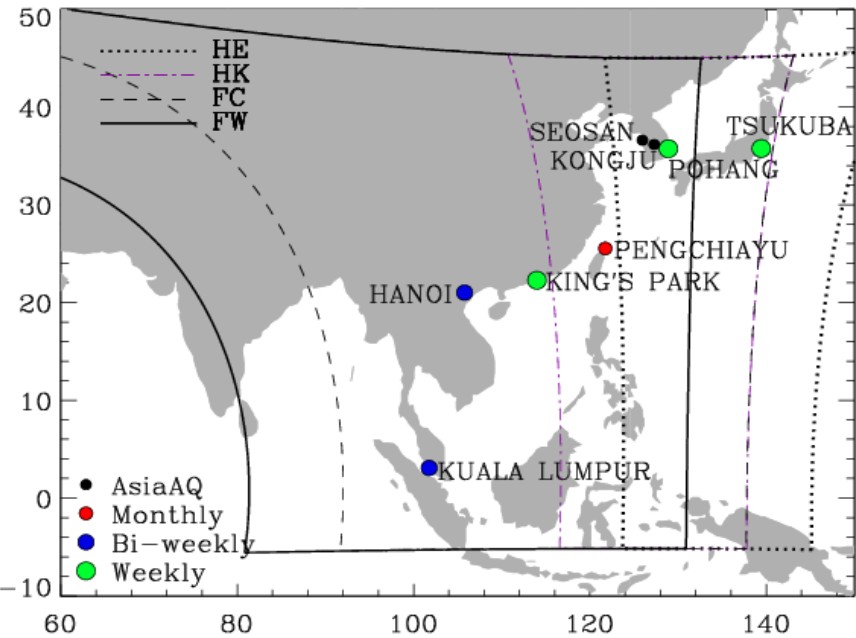

**Figure 1. Geographic coverage of the four GEMS scan modes: Half East (HE), Half Korea (HK), Full Central (FC),**
**and Full West (FW), indicated by the curved boundaries. Colored dots indicate ozonesonde stations with regular**
**launches within the GEMS domain, classified by launch frequency: red for monthly, blue for bi-weekly, and green for**
**weekly. Black dots represent additional sites that participated during the Asia-AQ campaign.**

## 2.2 Algorithm Heritage

The heritage of the ozone profile retrieval algorithm is rooted in long-standing achievements in developing, improving,
and validating ozone profile retrievals using satellite observation data from the Global Ozone Monitoring Experiment
(GOME), the OMI, the Ozone Mapping and Profiler Suite (OMPS), and the Tropospheric Monitoring Instrument (TROPOMI)
(Bak et al., 2017, 2024, 2025a; Cai et al., 2012; Dobber et al., 2008; Liu et al., 2005, 2010; Zhao et al., 2021). The Optimal
estimation technique (Rodgers, 2000) provides the foundation for solving the inverse problem, transforming spectral
information into geophysical data. The retrieval process iteratively optimizes the state to minimize the cost function that
accounts for both the differences between simulated and measured spectra and the deviation of the state vector from the a
priori vector. This optimization depends critically on stable wavelength and radiometric calibration, as well as an accurate





radiative transfer model, to ensure stable spectral fitting and reliable results. Algorithmic updates from the OMI Version 2.0
research product by Bak et al. (2024) were adopted in the development of the GEMS Version 3.0 ozone profile product. In
addition, calibration methodologies for GEMS L1C radiance and irradiance were newly implemented to ensure spectral fitting
stability and retrieval accuracy. The following sections provide a detailed description of the inversion framework and its
implementation.

## 2.3 Optimal Estimation

The Optimal Estimation-based inversion (Rodgers, 2000) is physically regularized toward minimizing the difference
between a measured spectrum $Y$ and a spectrum that is simulated by the forward model $F(X)$. Given an atmospheric state $X$,
the inversion is constrained by the measurement error covariance matrix $\mathbf{S_y}$ and statistically regularized by an a priori state
vector $X_a$ with a priori covariance matrix $\mathbf{S_a}$. The cost function (chi-square) and the updated equation for the posterior state
vector $X$ at iteration step $i + 1$ are written as

$$\chi^2 = \|\mathbf{S}_y^{-\frac{1}{2}}\{\mathbf{K}_i(X_{i+1} - X_i) - [Y - \mathbf{F}(X_i)]\})\|_2^2 + \|\mathbf{S}_a^{-\frac{1}{2}}(X_{i+1} - X_a)\|_2^2 \quad (1) \text{ and}$$

$$X_{i+1} = X_i + (\mathbf{K}_i^T \mathbf{S}_y^{-1} \mathbf{K}_i + \mathbf{S}_a^{-1})^{-1}[\mathbf{K}_i^T \mathbf{S}_y^{-1}(Y - \mathbf{F}(X_i)) - \mathbf{S}_a^{-1}(X_i - X_a)] \quad (2)$$

, where each component of the matrix $\mathbf{K}$ is the derivative of the forward model to the actual atmospheric state, called the
Jacobians or weighting function matrix.
The posterior error covariance matrix, quantifying the total uncertainty in the retrieved state $\hat{x}$, is given by:

$$\hat{\mathbf{S}} = \left(\mathbf{K}^T \mathbf{S}_y^{-1} \mathbf{K} + \mathbf{S}_a^{-1}\right)^{-1}. \quad (3)$$

The retrieval gain matrix $\mathbf{G}$, representing the sensitivity of the retrieval to the measurements, can be written as:

$$\mathbf{G} = \hat{\mathbf{S}} \mathbf{K}^T \mathbf{S}_y^{-1} \ (\mathbf{G} = \frac{\partial \hat{x}}{\partial y}). \quad (4)$$

The product of $\mathbf{G}$ and $\mathbf{K}$ then yields the averaging kernel matrix $\mathbf{A}$, which characterizes the sensitivity of the retrieved state to
the true atmospheric state:

$$\mathbf{A} = \mathbf{GK} \ (\mathbf{A} = \frac{\partial \hat{x}}{\partial x_{true}}). \quad (5)$$

Beyond information content analysis, the matrices $\mathbf{G}$ and $\mathbf{A}$ also govern the retrieval error characteristics. Accordingly, $\hat{x}$ can
be expressed as:

$$\hat{x} = \mathbf{A}x_{true} + (\mathbf{I}_n - \mathbf{A})x_a + \mathbf{G}\sigma_y \quad (6)$$





which represents a weighted combination of the true atmospheric state and a priori information, and adds the measurement
noise. The retrieval uncertainty due to measurement noise is quantified by propagating $\sigma_y$ from the measurement space into
the state space through the gain matrix $\mathbf{G}$, resulting into the measurement error covariance matrix:
$$\mathbf{S}_n = \mathbf{G}\mathbf{S}_y\mathbf{G}^T. \quad (7)$$

Meanwhile, the smoothing error covariance matrix, representing the retrieval uncertainty caused by limited vertical
information, is defined as:
$$\mathbf{S}_s = (\mathbf{A} - \mathbf{I})\mathbf{S}_a(\mathbf{A} - \mathbf{I})^T \quad (8)$$

These two contributions then add up to the total covariance as given in Eq. (3), or $\hat{\mathbf{S}} = (\mathbf{I} - \mathbf{A})\mathbf{S}_a$.

## 2.4 Implementation details

The state vector $\boldsymbol{X}$ includes 24 partial ozone columns, surface albedo ($0^{th}$ and $1^{st}$ order wavelength terms), cloud fraction,
and six additional calibration parameters (see Supplementary Table 3). The measurement vector $\boldsymbol{Y}$ consists of the logarithms
of the sun-normalized radiance spectra, which enhances retrieval stability by reducing the sensitivity to absolute radiance errors
and Fraunhofer lines. Measurement errors ($\sigma_y$) are assumed to be mutually uncorrelated. Since the GEMS L1C product does
not provide measurement error estimates, a constant relative error of 0.2% is uniformly applied across the spectral range.
Accordingly, the measurement error covariance matrix is defined as:
$\mathbf{S_y} = \text{diag}(\sigma_{y,1}^2, \sigma_{y,2}^2, ..., \sigma_{y,n}^2).$
Correlations between ozone layers are accounted for using a correlation length $L$ of 6 km in the a priori error covariance
matrix, defined as:
$\mathbf{S_a} = \sigma_i^a \sigma_j^a exp(-|i - j|/L),$
where $\sigma_i^a$ and $\sigma_j^a$ are the a priori errors of the $i^{th}$ and $j^{th}$ state vector components, respectively. The updates from GEMS v2.0
to v3.0 mirror those from OMI v1.0 to v2.0. In particular, the radiative transfer model is replaced with the PCA-VLIDORT
v2.6 (Bak et al., 2021) to enhance the simulation efficiency. A look-up table correction was also newly implemented to account
for approximations in the radiative transfer calculation related to the number of streams, coarse vertical layering, and
polarization treatment. The TSIS-1 Hybrid Solar Reference Spectrum (Coddington et al., 2021) is now used instead of the
solar reference from Chance and Kurucz ( 2010). The ozone cross-section has been changed from BDM 1995 (Brion et al.,
1993; Daumont et al., 1992; Malicet et al., 1995) to BW (Birk and Wagner, 2018). Notably, the a priori ozone profile, based
on the tropopause-based ozone climatology (Bak et al., 2013), has been consistently used in GEMS v2.0, GEMS v3.0, and
OMI v2.0. The temperature data are necessary to account for the temperature dependence of the ozone cross-section, while
surface and tropopause pressures are used to define the 25-level pressure grids (Supplementary Fig. 1). The tropopause pressure
is also used to convert the a priori ozone profile from a tropopause-based to a surface-based vertical coordinate system. For
meteorological inputs, the Global Forecast System (GFS) of a National Centers for Environmental Prediction (NCEP) weather



forecast model is used in the daytime processing (DRPO) mode. GFS data are downloaded daily at 05:00 KST, covering
forecast periods between 6 KST and 18 KST, with lead times of 12 to 21 hours. In the reprocessing (RPRO) mode, the
meteorological input is switched to the NCEP FNL (Final) Operational Global Analysis data. The meteorological fields,
provided at 3-hour intervals (GFS) or 6-hour intervals (FNL) per day, are interpolated to match the GEMS reference time
(HH:45).

## 2.5 Calibration methodologies

The calibration process consists of several key components: on-orbit slit function derivation and wavelength calibration
to ensure spectral accuracy (Section 2.5.1), as well as irradiance offset correction and soft calibration to reduce radiometric
uncertainties (Section 2.5.2).
### 2.5.1 spectral correction

The instrument spectral response function (ISRF), or slit function is required to degrade high-resolution spectra (e.g.,
absorption cross-sections) to match the spectral resolution of GEMS. Pre-flight ISRFs, measured at six discrete wavelengths
and interpolated across all 1,322 wavelength grids, are available (Kang et al., 2022). However, our companion study proposes
an on-orbit slit function derivation for GEMS based on a super-Gaussian model to account for temporal variations in the
instrument response (Bak et al. 2025b), and is therefore not repeated here. That study also reports that the irradiance spectrum
should be shifted by 0.055 nm to align with the Fraunhofer lines. Most GEMS Level 2 trace gas algorithms assume similar
spectral shifts for radiance and irradiance, applying the irradiance-derived shift directly to the radiance spectra. However,
Figure 2 reveals substantial discrepancies in both the magnitude and spatial pattern of the spectral shift between radiance and
irradiance, ranging from 0.02 to 0.04 nm, with larger differences observed toward the northern edge of the spatial domain. To
ensure computational efficiency in operational processing, the radiance shift is determined from the first mirror step and
applied uniformly along the scan direction, based on the observation that spectral shifts in the radiance data remain relatively
uniform across mirror steps. Additionally, as degradation progresses, pixel-to-pixel perturbations increase toward the central
spatial pixels in both radiance and irradiance measurements.



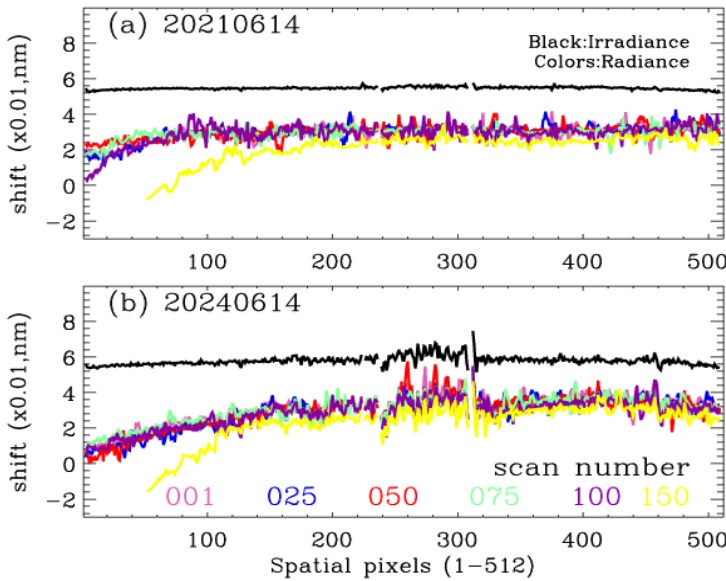

**Figure 2. Shifts of irradiance and radiance relative to the solar reference from Coddington et al. (2021), shown as a function of spatial pixel number (1–512) for (a) June 14, 2021 (20210614) and (b) June 14, 2024 (20240614). Colored lines represent the scan lines (mirror steps) plotted at 25-intervals, ranging from 1 to 150.**

### 2.5.2 Radiometric correction

The GEMS irradiance is spatially and seasonally biased due to a missing calibration component for the BTDF, which defines how light transmits through a diffuser based on incident and outgoing angles—a well-known issue (Kang et al. 2024; Bak et al. 2025b). Additionally, Bak et al. (2025b) identified progressive radiometric degradation, resulting in an annual irradiance decrease of ~5% in the shorter UV range. They also reported that the measured irradiance is roughly 40% lower than the solar reference near 325 nm. Because normalized radiance is used in spectral fitting, such irradiance biases can directly propagate into retrieval output. To address these discrepancies, a major revision was implemented in version 3. Specifically, a scaling correction factor was introduced to compensate for the systematic difference between the GEMS irradiance ($I_m$) and a high-resolution solar reference spectrum ($I_{ref}$). This correction factor (C) is derived by minimizing the following cost function:

$$\chi^2 = \sum_\lambda \left( I_m(\lambda) - \left[ C \cdot I_{ref}(\lambda + \triangle \lambda) \otimes S + \sum_m^3 P_b^m \left(\lambda - \bar{\lambda}\right)^m \right] \right)^2 \quad (9)$$

where:

- S: instrument spectral response function (ISRF)
- $\otimes$: convolution operator,



- $\triangle \lambda$: wavelength shift
- $P_b^m$: coefficients of a third-order baseline polynomial centered at $\bar{\lambda}$

In this approach, the slit function parameters and the wavelength shift are first determined independently and then held fixed, allowing the solar reference spectrum to be adjusted to the measured irradiance in terms of spectral resolution and spectral alignment. The scaling factor $C$ and the baseline polynomial $P_b$ are subsequently fitted to capture remaining radiometric differences. As presented in Figure 3, the derived values of $C$ exhibit significant seasonal and spatial variations in irradiance offset related to angular dependence, along with a gradual temporal decline attributable to optical component degradation, particularly at the middle spatial pixels. In version 3, only the scaling factor $C$ is applied in the irradiance correction, by dividing the irradiance by C. This decision was made because applying the baseline polynomial $P_b$ directly to the irradiance introduced artificial structures into the spectral fitting of the normalized radiance, resulting in a significant underestimation of stratospheric ozone retrievals. Residual wavelength-dependent uncertainties are instead addressed through the soft calibration process, which has been newly implemented in version 3. This empirical correction eliminates systematic biases in the normalized radiance by applying adjustment factors derived from the ratio of measurements to simulated spectra based on accurate forward model calculations. The ozone profile input for the forward model calculation is constructed using daily zonal mean Microwave Limb Sounder (MLS) data (Livesey et al., 2025) above 215 hPa and climatological profiles (McPeters and Labow, 2012) below that level, with the integrated total column adjusted to match the zonal mean total ozone from daily OMPS measurements. A one-week set of clear sky measurements, collected at 02:45 UTC between July 11 and 17, 2021, is used to derive the soft calibration spectra as a function of the 512 spatial pixels. While a cloud fraction threshold of 0.2 is typically used to define clear-sky conditions, we relaxed this criterion to 0.4 due to the known overestimation in the GEMS cloud product, which is also affected by irradiance offsets. Figure 4 illustrates the derived soft spectra and the impact of applying the irradiance correction. After correction, the soft calibration spectra show significantly reduced biases and improved spatial consistency. The residual biases are generally positive and remain below 3% for most pixels, except for a few central pixels that exhibit negative values, possibly due to unflagged dead pixels in the GEMS L1C data. In contrast, without the correction, substantial wavelength-and spatially dependent biases are evident, with systematic biases ranging from 3% to 10% in the shorter UV range. Moreover, the standard deviation of the residual spectra stays below 1% for spatial pixels numbered below 100, while it increases above 3% for pixels above 400 without correction. With correction applied, this increase is limited to 2 %. Figure 5 demonstrates the resulting improvement in spectral fitting accuracy achieved through the application of both radiometric (scaling correction to irradiance and soft calibration to normalized radiance) and wavelength calibration in version 3, compared to version 2. With these corrections, mean fitting residuals decreased from approximately 0.8% in v2.0 to 0.2% in v3.0 across most spatial pixels, representing more than a fourfold enhancement in retrieval precision. Version 3.0 not only reduces the mean fitting residuals but also achieves substantial improvements in seasonal stability, spatial uniformity,





and the removal of systematic and random artifacts, —highlighting the effectiveness of the enhanced calibration and retrieval

procedures.

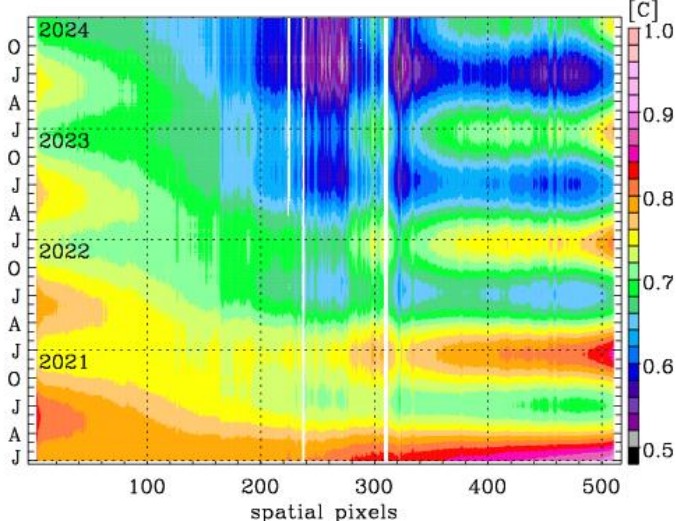

**Figure 3. Time–space distribution of the derived scaling correction factor C across 512 spatial pixels from January 2021 to December 2024. The scaling factor C, fitted over the 310–330 nm spectral window, represents the ratio between GEMS irradiance and a high-resolution solar reference spectrum.**




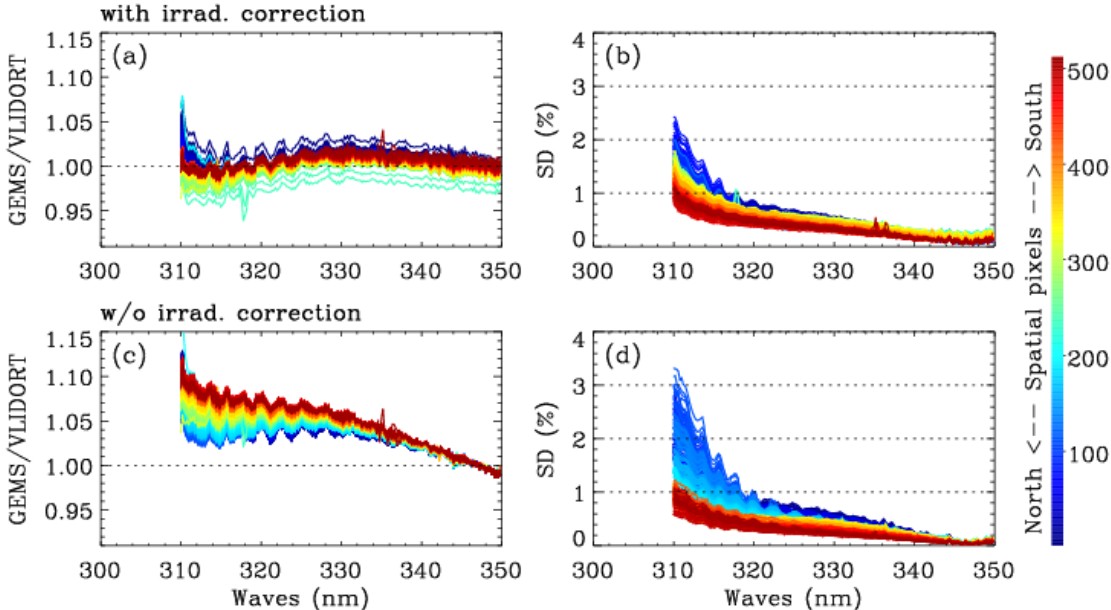

**Figure 4. GEMS soft spectrum, derived as the mean difference between measured and simulated normalized radiances, as a function of wavelength (300–350 nm) for each of the 512 spatial pixels (color-coded from north to south), with the standard deviation of the mean difference. The upper panel includes the scaling correction for the irradiance offset, while the bottom panel does not.**

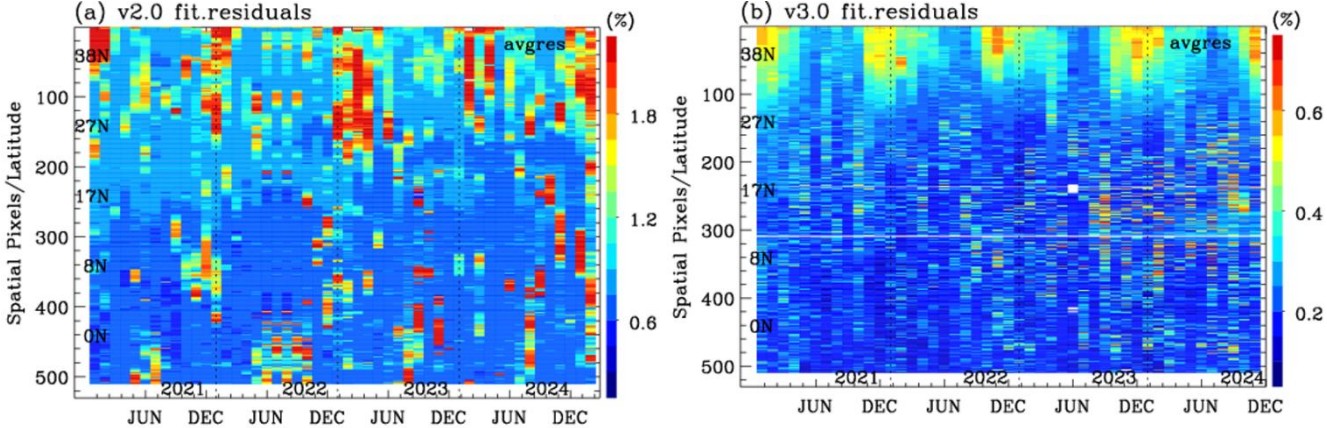

**Figure 5. Comparison of spectral fitting quality from ozone profile retrievals between versions 2.0 and 3.0, averaged over the first 20 scanlines and shown as a function of the 512 spatial pixels. The evaluation is performed on the 15th day of each month from 2021 to 2024 (04:45 UTC). Fitting residuals, calculated as the root mean square (RMS) of the relative differences between measured and simulated radiance (%), are stored as "ResidualOffit" in version 2 and "avg_residuals" in version 3. Note that the color scale range in panel (b) is narrowed to one-third of panel (a) to enhance the visibility of lower residual values.**



### 3. Retrieval Characterization

The retrieved ozone profiles can be characterized by their information content and associated uncertainties, assessed using the averaging kernel matrix (AKM) and error covariance matrix (CVM) for each profile. These characteristics are primarily influenced by the choice of fitting window, measurement noise, and the a priori covariance matrix, and remain largely unchanged between version 2 and version 3 of the retrieval algorithm. The rows of the AKM serve as vertical smoothing functions, indicating the sensitivity of the retrieved ozone concentration to changes in the true atmospheric state (see Eq. 5). The trace of the AKM yields the degrees of freedom in the signal (DFS), representing the total number of independent pieces of information in the retrieval. DFS can also be calculated for specific vertical sub-columns using partial traces. Retrieval uncertainty, given by the square root of the CVM diagonal, is assessed relative to the a priori uncertainty, both in terms of total uncertainties and the contribution from measurement noise alone.

Figure 6 shows average averaging kernels and uncertainty profiles from the GEMS 04:45 UTC scan on 15 June and 16 December 2021, for two observation locations with different optical paths determined by solar zenith angles (SZA) and viewing zenith angles (VZA). The retrieval typically yields the most information (highest averaging kernel peaks) where the a priori uncertainty is highest, notably just below the stratospheric ozone layer and in the upper troposphere and lower stratosphere (UTLS). The location of these maxima, however, strongly depends on the optical path length (SZA and VZA). At the kernel peaks, retrieval uncertainty is reduced by ~ 50 % with respect to the a priori, with about one-third of the solution error being due to measurement noise. Away from the peaks, the reduction in uncertainty is generally smaller. At high SZAs, negative kernel oscillations are evident, indicating challenges in vertically allocating the measurement information. This suggest that the retrieval may offer limited improvement over the prior under such conditions.

Figure 7 presents the sub-columns DFS values for the troposphere and stratosphere, based on the same observation cases shown in Figure 6. The corresponding ozone partial columns are also included (Supplementary Fig. 2), given the expected dependence of the information content on the atmospheric ozone concentration. The stratospheric DFS clearly increases with optical path lengths, and thus with latitude, especially winter (Fig. 7d). In the troposphere, the effect of optical path length is weaker and only apparent at higher latitudes, showing the opposite behaviour: DFS values decrease with increasing SZA and VZA (Fig. 7c). (Fig. 7c). Additionally, there is a strong correlation with the tropospheric ozone burden: the retrieval yields more information when ozone concentrations are higher, resulting in stronger absorption signals. Adding up the tropospheric and stratospheric DFS contributions, the total DFS typically ranges from 1.5 to 3, with a somewhat compensating effect between tropospheric decreases and stratospheric increases for the higher latitudes.




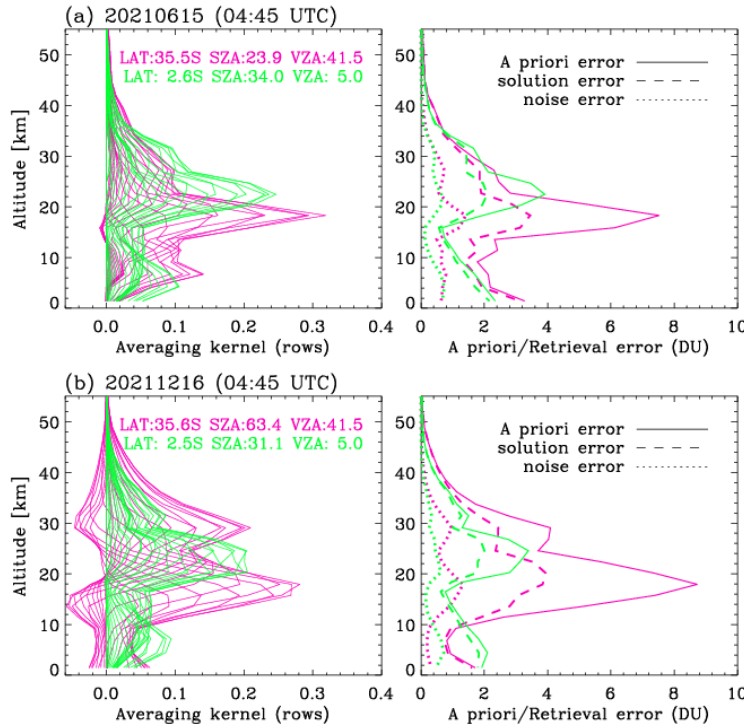

**Figure 6. Averaging kernels and retrieval errors of ozone profiles from GEMS on 15 June (a, top) and 16 December (b, bottom) 2021. The pink and green lines represent averages over cross-track pixels 50-100 and 450-550, respectively, at the first scan line.**



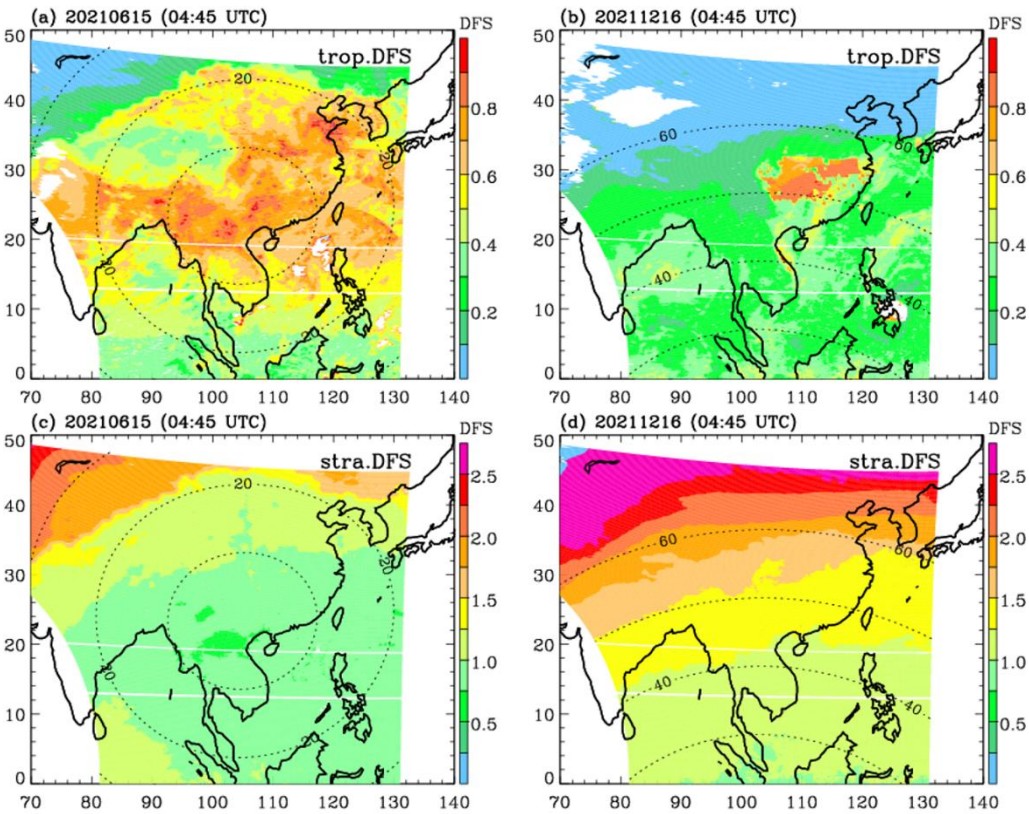

**Figure 7. Degrees of freedom for signal (DFS) for tropospheric column ozone on (a) 15 June and (b) 16 December 2021, and analogous for the stratospheric column in (c) and (d), respectively. Contours indicate the solar zenith angle (SZA) at 20° intervals. The corresponding ozone distributions are shown in Supplementary Figure 2.**

In this work, we further examine three supplementary diagnostics: the retrieval sensitivity, retrieval offset and effective vertical resolution (which differs from the sampling resolution), following Keppens et al. (2015). The sum of each row of the AKM quantifies the total retrieval sensitivity, providing a vertically resolved and normalized measure of the contribution from satellite observations relative to the a priori profile. The retrieval offset indicates any vertical mismatch between the location of maximum sensitivity (the retrieval barycenter) and the nominal retrieval altitude (Rodgers, 2000). The effective vertical resolution is derived from the width of each averaging kernel, treated as a vertical smoothing function. Here, we use the full width at half maximum (FWHM) as a measure of vertical resolution. This measure, however, does not consider averaging kernel oscillations, including the presence of negative values (see Figure 6). Figure 8 presents the retrieval diagnostics described above—sensitivity, offset, and vertical resolution—for GEMS ozone profile retrievals, evaluated from every tenth mirror step and spatial pixel, resulting in the order of 1000 profiles per daily plot (89088/100 for the 04:45 UTC scan). SZA and VZA are major quantities affecting the retrieval performance, while other influence quantities are examined in the Supplement. The results show that, on average, the vertical sensitivity of the ozone profile retrievals is close to unity throughout





most of the profile. Sensitivity decrease to values below 0.5 only in the lowest 5 km, with higher values occurring above highly
reflective surfaces, including high cloud fractions. As expected, sensitivity drops significantly below clouds. In the troposphere,
vertical sensitivity generally increases with decreasing path length (e.g., lower SZA and VZA), as shorter paths enhance
atmospheric penetration. In the stratosphere, however, higher sensitivities are observed for more oblique viewing geometries
(higher VZA), particularly during winter when SZAs are also high, resulting in increased DFS. In contrast, during summer,
when SZAs are lower, stratospheric sensitivity is higher for near-nadir viewing angles (i.e., shorter path lengths).
Outside of the UTLS (about 15-30 km), the retrieval barycenter deviates approximately linearly from the nominal retrieval
altitude. This means that the vertical sensitivity is primarily distributed off-diagonal, with its barycenter located in the UTLS
(as can also be seen from the averaging kernel peak positions in Figure 6), and results in a rather low average retrieval DFS
(also see the sum of both partial DFS values in Figure 7). The retrieval offset depends on SZA and VZA (and hence latitude)
in the troposphere but shows no other significant dependences on the influence quantities under study. The offset is reduced
for more sideways solar irradiance and observation of the troposphere, although it has to be taken into account that the
tropospheric retrieval sensitivity is at the same time reduced as well (see above).
The average effective vertical resolution of the GEMS ozone profiles ranges from 6 to 10 km, with the highest values
founded in the lower stratosphere. Both in the troposphere and stratosphere, actual values again strongly depend on SZA and
VZA, resulting in a meridian dependence as well, but again an opposite behavior is observed above and below the tropopause:
Longer path lengths result in coarser vertical resolutions (reduced FWHM) in the troposphere, while the opposite happens in
the stratosphere, especially for very high SZA during winter, meaning the retrieved information is distributed over a larger
vertical extent.



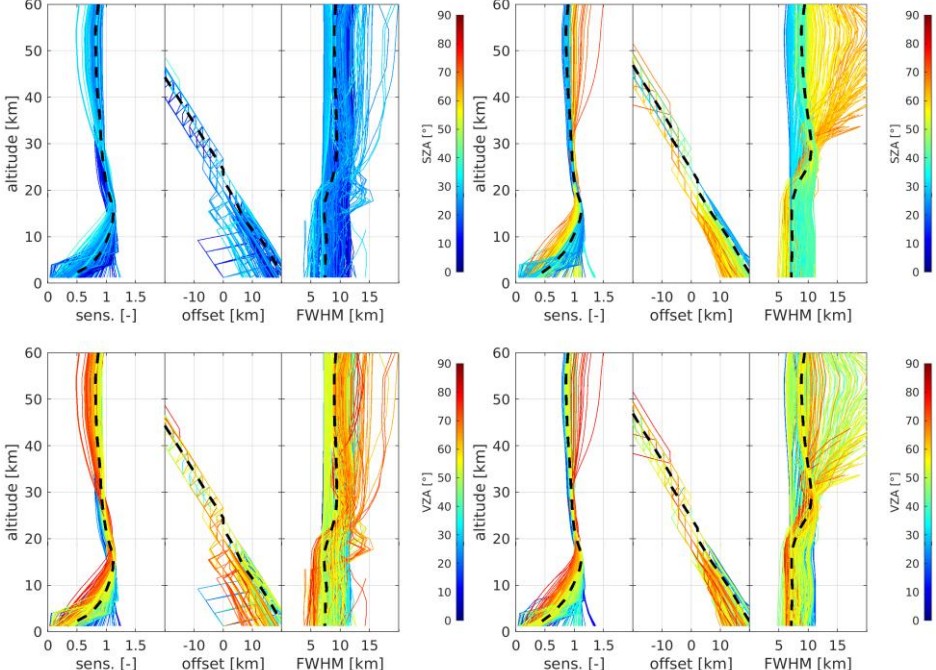

**Figure 8: GEMS ozone profile retrieval information content in terms of sensitivity, offset, and kernel FWHM for June 15 (left) and December 16 (right) 2021, and for SZA (top) and VZA (bottom) as physical influence quantities, color-coded in each plot. Median values are indicated by black dashed lines.**

## 4. Validation using independent reference datasets

As a preliminary step in establishing a reliable validation framework for GEMS ozone profile retrievals, Bak et al. (2019) evaluated ozonesonde soundings from 10 East Asian sites and found that electrochemical concentration cell (ECC) sensors provided more reliable measurements than modified Brewer–Mast (MBM) and carbon–iodine (CI) sondes. They also emphasized the importance of consistent procedures across preparation, operation, and post-processing stages to ensure the long-term consistency of data quality. Among these sites, five—Pohang, Hong Kong, Tsukuba, Hanoi, and Kuala Lumpur—have remained active during the GEMS mission, regularly launching balloon-borne ECC ozone sensors. Weekly regular observations have continued at Pohang, King's Park, and Tsukuba in the afternoon (1:30-2:30 pm LT). While Hanoi and Kuala Lumpur provide bi-weekly observations, they were not recommended as reference sites in Bak et al. (2019) due to frequent changes in either the sensing solution concentrations or the ozonesonde manufacturer. However, these inconsistencies have been better managed during the GEMS operational period, and thus the data from these sites are included in this study. In addition, monthly ozonesonde observations from Pengchiayu, which began in 2022, are also incorporated. Table 1 summarizes the availability of the regular ozonesonde sites used for GEMS validation in Section 4.1. In Section 4.2, we additionally use a total of 13 ECC ozonesondes launched at Seosan (126.38°E, 36.92°N) and 10 launched in March at Kongju (127.74°E,



36.47°N), South Korea, as part of the 2024 Airborne and Satellite Investigation of Asian Air Quality campaign (NASA, 2023).
These two sites, approximately 131 km apart, are marked by black symbols in Figure 1. Ozonesonde measurements can be
spatially matched with GEMS FW observations taken at 04:45 UTC, except for those from the Tsukuba station, which lies
outside the FW domain. Instead, the Tsukuba station falls within the coverage of the FC scan, which operates at 01:45, 02:45,
or 03:45 UTC, depending on the season. Integrated total ozone columns were also evaluated using Pandora measurements
(Herman et al., 2015) at Seosan during the ASIA-AQ campaign (Section 4.2).

**Table 1. List of regular ozonesonde stations used in this study.**

| Station | Pohang | Tsukuba | King's park | Hanoi | Pengchiayu | Kuala Lumpur |
|---|---|---|---|---|---|---|
| **Country** | South Korea | Japan | Hong Kong | Vietnam | Taipei | Malaysia |
| **Lon, Lon (deg.)** | 36.03, 129.38 | 36.06, 140.13 | 22.31, 114.17 | 21.02, 105.804 | 25.63, 122.08 | 2.73, 101.7 |
| **Provider[#]** | KMA | WOUDC | WOUDC | SHADOZ | WOUDC | SHADOZ |
| **Frequency** | Weekly | Weekly | Weekly | Bi-weekly | Monthly | Bi-weekly |
| **Launch Time (LT)** | 2:00 pm | 2:30 pm | 1:30 pm | 1:00 pm | 12:00 am | 12: 30 am |
| **Beginning date** | 1995-01-12 | 2017-06-22 | 2000-01-05 | 2004-09-18 | 2022-04-18 | 1998-05-04 |
| **Latest update** | 2024-06-26 | 2025-02-27 | 2024-12-31 | 2024-02-23 | 2024-05-12 | 2022-12-22 |

[#]KMA (Korea Meteorological Administration), WOUDC (World Ozone and Ultraviolet Radiation Data Centre), SHADOZ
(Southern Hemisphere ADditional OZonesondes)







**4.1 Validation with regular ozonesonde soundings**

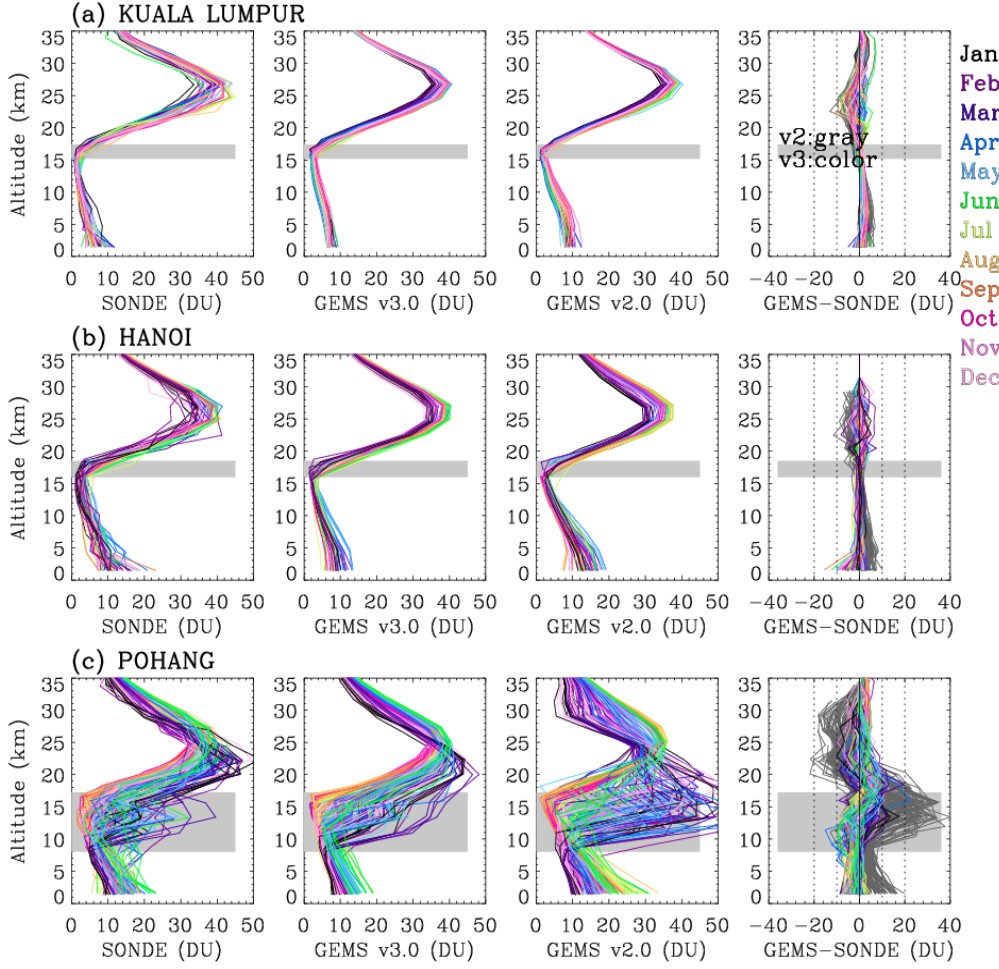


**Figure 9. Ozone vertical profiles (in DU) at three sites—(a) Kuala Lumpur, (b) Hanoi, and (c) Pohang—during 2021–**
**2024. Each panel displays individual ozonesonde soundings along with corresponding GEMS v3.0 and v2.0 retrievals.**
**The last columns present the respective differences (GEMS − SONDE) for GEMS v3.0 and v2.0, displayed in color and**
**dark gray, respectively. The gray shaded area denotes the range of tropopause altitudes (minimum to maximum).**

Figure 9 illustrates how well GEMS captures the vertical distribution of ozone up to 35 km—the typical burst altitude of
ozonesonde balloons—at three stations representing different latitudinal regions: mid-latitudes (Pohang), subtropics (Hanoi),
and tropics (Kuala Lumpur). Compared to the previous version, the updated GEMS v3.0 demonstrates substantial
improvements in reproducing ozone vertical profiles, particularly at the mid-latitude site of Pohang. In GEMS v2.0,
tropospheric ozone was overestimated by up to 20 DU in the lower troposphere and by as much as 40 DU in the tropopause
region. Additionally, stratospheric ozone columns were underestimated by up to 20 DU relative to ozonesonde measurements.



These discrepancies are notably mitigated in GEMS v3.0, with tropospheric biases reduced to within 10 DU and stratospheric
biases to within 5 DU. At lower latitude sites, both GEMS v2.0 and v3.0 produce qualitatively similar ozone profiles, as the
vertical structure shows relatively weak seasonal variability and the tropopause altitude remains stable, making it easier to
constrain with a priori information. The retrieved ozone profiles from satellite nadir-view observations generally exhibit weak
vertical sensitivity, particularly in the lower troposphere. The use of integrated column ozone is recommended to enhance the
information content, as it allows for a more practical validation compared to profile-based approaches. Figure 10 presents time
series comparisons of lower tropospheric ozone columns (below 300 hPa) derived from GEMS (v2.0 in gray and v3.0 in black)
and ozonesonde observations (in red) at six stations across different latitudes during the period 2021 to 2024.

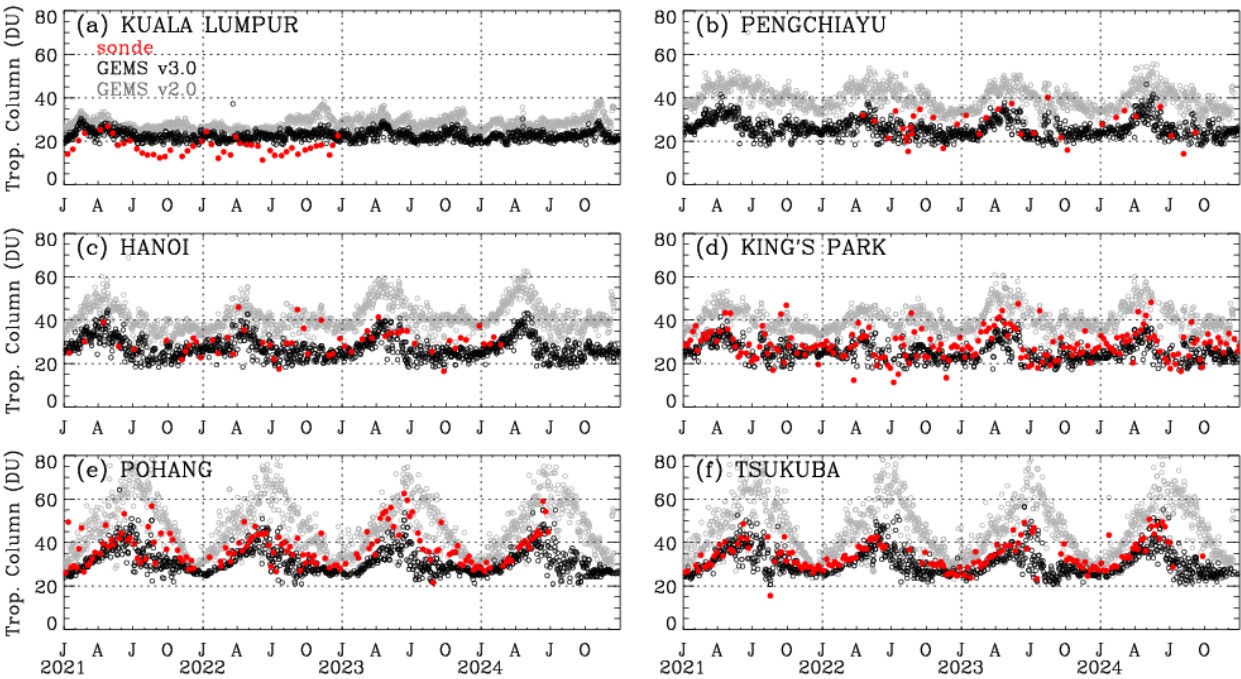


**Figure 10. Time-series of tropospheric ozone columns from GEMS v3.0 (black), GEMS v2.0 (grey), and ozonesondes**
**(red). The x-axis marks the months of the year using initials: J (January), A (April), J (July), and O (October).**

Mid-latitude sites (Pohang and Tsukuba) exhibit pronounced seasonality, with ozonesonde-derived tropospheric ozone
columns ranging from 25 to 50 DU—peaking in summer and declining toward winter. Within the summer season, ozone levels
typically reach their maximum in June, followed by a sharp decline in July and August. As shown, GEMS v3.0 reasonably
reproduces this seasonal pattern. At subtropical sites such as Hanoi, King's Park, and Pengchiayu, seasonal changes are less
pronounced, with ozone columns typically fluctuating between 20 and 45 DU. A distinct spring peak of 40-45 DU is
consistently observed in both ozonesonde and GEMS v3.0 time-series. The lowest ozone levels are observed between July and
October, remaining a few DU lower than the wintertime minimum. At the tropical site of Kuala Lumpur, ozonesonde



measurements are limited in 2021 and 2022, but the available data suggest minimal seasonal variation in tropospheric ozone,
consistent with the weak seasonal signals typically observed in the tropics. With its dense temporal coverage, GEMS v3.0
complements the sparse ozonesonde measurements and identifies the flat tropospheric ozone levels throughout the 2021-2024
period. However, GEMS v2.0 systematically retrieves higher ozone levels across all latitudinal bands. This overestimation is
much more pronounced at mid-latitudes than at lower latitudes. In particular, GEMS v2.0 significantly overestimates summer
ozone values by 30 DU compared to GEMS v3.0, with the discrepancy decreasing toward winter. In the subtropics, the
difference between GEMS v2.0 and v3.0 remains about 15 DU, without clear seasonal change. In particular, GEMS v2.0
retrieves higher ozone amounts in 2023 and 2024 compared to earlier years, which is not reflected in either GEMS v3.0 or
ozonesonde data. This increasing discrepancy is likely associated with the optical degradation of the instrument, which leads
to decreasing irradiance values over time (Kang et al. 2024; Bak et al. 2025b) and, in turn, affects the accuracy of the ozone
profile retrievals. In the tropics, the GEMS products from both versions agree within 5 DU during 2021–2022, but the
difference increases to within 10 DU in 2023–2024. Notably, the issues identified in GEMS v2.0 are substantially mitigated
in GEMS v3.0, owing to the newly implemented radiometric calibration applied to both irradiance and normalized radiance.

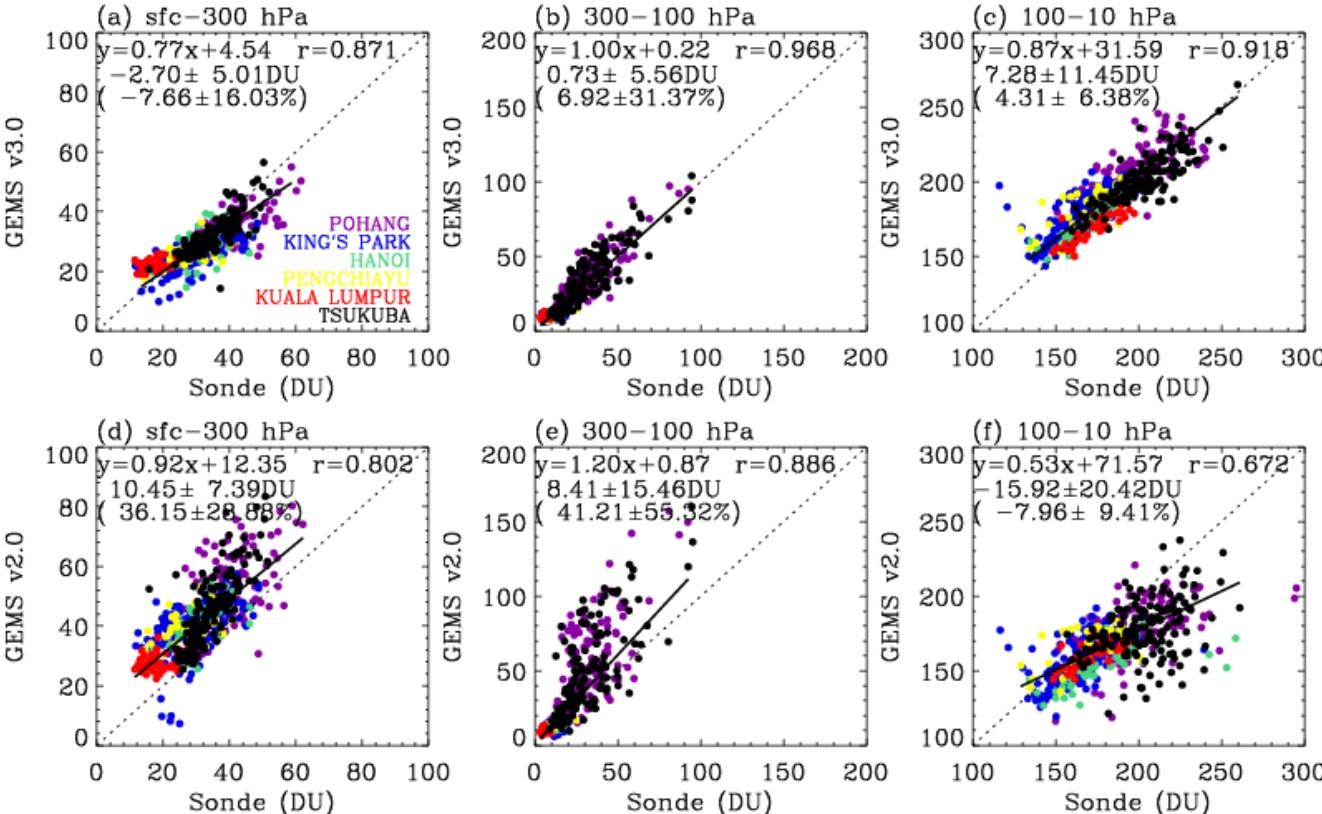


**Figure 11. Scatter plots of GEMS and ozonesonde ozone columns for three different layers, surface-300 hPa, 300-100**
**hPa, and 100-10 hPa. The upper (a-c) panels show results from GEMS v3.0, and the bottom panels (d-f) from GEMS**



**v2.0. Each data pair is color-coded by station. Regression lines and correlation coefficients (r) are derived from all data pairs, along with the mean bias and standard deviation reported in both DU and percentage.**

The quantitative comparison between GEMS and ozonesonde measurements is presented in Figure 11 (a,d) for tropospheric ozone columns below 300 hPa. Compared to version 2.0, which exhibits a substantial positive bias of 36.15% and high variability (±28.88%), GEMS v3.0 shows a marked improvement, reducing the bias to −7.66% with lower scatter (±16.03%) and achieving a higher correlation with ozonesonde observations (r = 0.87 vs. r = 0.80). The regression slope for GEMS v2.0 is closer to unity than that of v3.0, due to the presence of both negative biases at high-ozone sites and positive biases at low-ozone sites, whereas v2.0 shows more uniform positive biases across stations. Figure 11 also evaluates ozone partial columns in upper troposphere and lower stratosphere (UTLS: 300-100 hPa) and the middle stratosphere (100-10 hPa), respectively. The 100–10 hPa layer, which corresponds to the ozone maximum in the upper stratosphere, also shows good agreement, with GEMS v3.0 achieving a correlation of r = 0.92 and a relatively small mean bias (4.31 ± 6.38 %), further supporting the reliability of the updated retrievals at higher altitudes.

The GEMS retrievals are inherently more influenced by a priori information compared to existing nadir satellite products such as OMI and TROPOMI, due to the narrower spectral range (310-330 nm versus 270-330 nm). Despite both versions employing the same a priori constraints, GEMS v2.0 exhibits poorer agreement with ozonesonde data than the a priori itself, reflecting the detrimental impact of radiometric uncertainties on the retrievals. However, GEMS v3.0 demonstrates better agreement than the a priori, indicating improved retrieval performance, especially when retrieving high ozone concentrations in both the troposphere and stratosphere. A comparison between GEMS a priori and ozonesondes is provided in Supplement Figure 4.

## 4.2 Validation with Asia-AQ campaign ozonesonde soundings

In Figure 12, the ozonesonde measurements are presented as ozone mixing ratio profiles. Ozone concentrations near the surface range from 30 to 50 ppb, which are lower than those in the upper troposphere—approximately 60 ppb in February and increasing to 80 ppb in March. These observed tropospheric ozone structures and their temporal variations are consistently reproduced from GEMS v3.0 retrievals. Notably, during the ozonesonde data gap in early March, GEMS v3.0 provides valuable supplementary information, revealing a downward propagation of ozone-rich air from the upper to the lower troposphere over time. Above the tropopause (~ 10 km), ozone mixing ratios generally exceed 0.1 ppm. The superimposed potential temperature profiles remained temporally stable in the stratosphere, reflecting persistent stratification and limited vertical dynamical activity. However, ozone mixing ratios in the lower stratosphere, particularly below 15 km, exhibited marked variability between 0.3 and 0.5 ppm, likely associated with isentropic transport. GEMS v3.0 effectively captures these



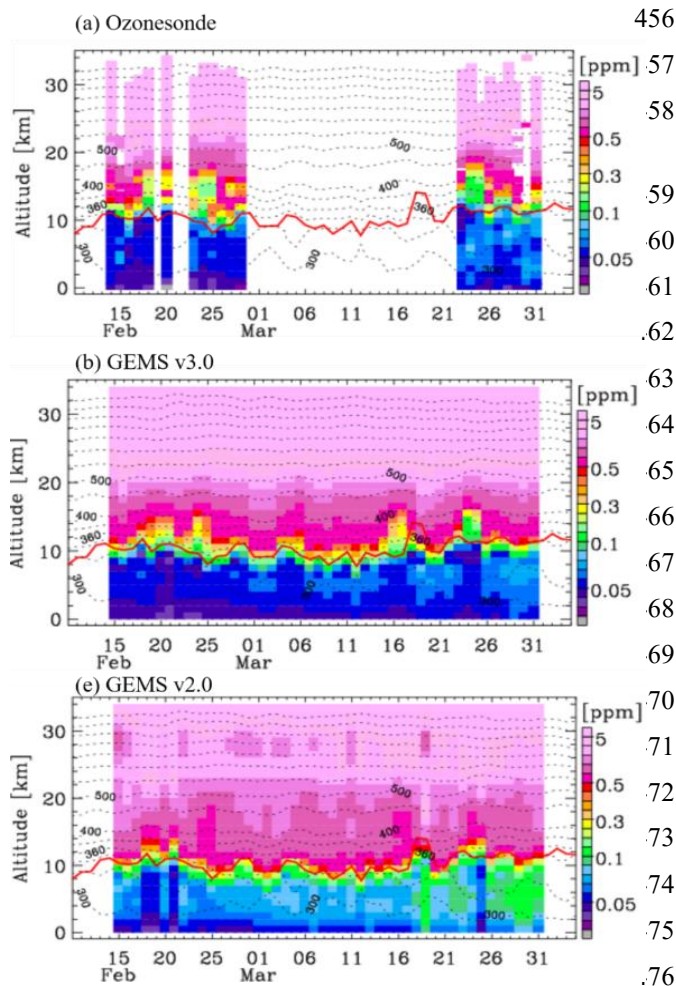

variations, demonstrating enhanced ability to resolve ozone fluctuations in the lower stratosphere compared to v2.0, which shows smoother, less structured patterns.

The evaluation of integrated ozone profiles as total ozone can provide useful insight into the overall accuracy and consistency of vertical profile retrievals when compared with well-established ground-based total column measurements (Bak et al., 2015). Ground-based Pandora total ozone column measurements at Seosan are used as a reference to evaluate the total ozone integrated from GEMS ozone profiles during the Asia-AQ campaign (Figure 13). An intercomparison of total ozone columns from GEMS (Baek et al., 2023), OMPS (Jaross, 2017), and TROPOMI (Garane et al., 2019) is also included to assess the consistency between GEMS ozone products ($O_3P$ and $O_3T$) and to evaluate the relative performance of GEMS compared to other satellite observations. As shown, total ozone values recorded by Pandora ranged from 300 to 450 DU during February and March 2024. These records closely align with satellite observations, evidenced by correlation coefficients of 0.97 or higher across all products. However, the retrievals from GEMS $O_3T$ show inconsistent performance between low and high ozone levels, resulting in a regression slope of 0.9, whereas the other satellite products exhibit slopes close to unity. GEMS $O_3T$ also significantly underestimates Pandora measurements, with a mean bias of –20 DU, primarily due to uncertainties in irradiance calibration (Baek et al., 2023). The scatter in the OMPS total ozone comparison is larger than that of the other products—by a factor of two—likely due to its coarse spatial resolution. GEMS $O_3P$ shows better agreement than the other satellite products, both in terms of scatter and biases, with mean differences ranging from 1.5 to 8 DU (-3.66 ± 4.27 DU).

**Figure 12. Time series of daily ozone mixing ratio profiles from ozonesondes and GEMS (v3.0 and v2.0) during the 2024 Asia-AQ campaign. The red line denotes the thermal tropopause, while the black contour lines (at 50 K intervals) represent potential temperatures, derived from the FNL meteorological product.**

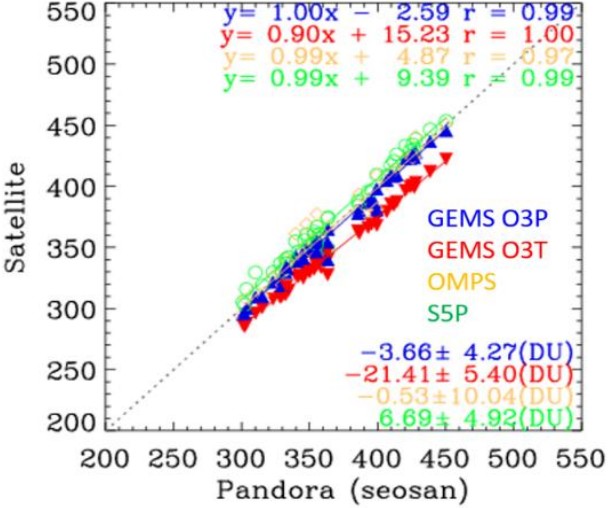

**Figure 13. Scatter plots of total ozone columns retrieved from satellite observations (GEMS O₃P, GEMS O₃T, OMPS, and TROPOMI) against Pandora measurements at Seosan during February–March 2024. Regression lines (y=slope·x+ intercept) and correlation coefficients (r) are shown in the top legend, while the bottom legend presents mean bias ± 1σ for each product. For comparison, Pandora observations are averaged within ± 30 min of 04:45 UTC each day and the satellite-Pandora pairs are selected based on the nearest satellite pixel located within 100 km of the Pandora site.**

## 5.  Conclusions for Version 3 and Remarks for the Next Version

This study provides the first detailed description of the GEMS operational ozone profile retrieval algorithm in the literature, along with an analysis of its retrieval characteristics in the 310–330 nm spectral range. The vertical sensitivity of the GEMS ozone profile is close to unity throughout most of the atmosphere. A decrease to values below 0.5 is observed only in the lowest five km. Outside of the lower stratosphere (about 15-30 km), the vertical sensitivity is mostly found off-diagonal, resulting in a rather low average retrieval DFS of about 1.5, up to 3 at maximum. The effective vertical resolution of the GEMS O3P retrieval amounts to 5-10 km.

This work primarily highlights substantial algorithmic and calibration enhancements implemented in version 3.0 over the previous version. Unlike other Level 2 algorithms that typically assume a uniform spectral shift, this work accounts for independent spectral shifts in radiance and irradiance. To address significant irradiance offsets—arising spatially and seasonally from BTDF-induced effects, and temporally from optical component degradation—a scaling factor correction is introduced. This scaling factor basically represents the ratio between the measured irradiance and the solar reference, capturing systematic deviations due to calibration limitations. Additionally, a soft calibration is applied to compensate for residual wavelength-dependent uncertainties not addressed by the scaling factor, as well as for spatial (cross-track) variations in



normalized radiance. The GEMS soft spectra are derived from clear-sky observations during the week of July 11–17, 2021,
at 02:45 UTC, to address systematic residuals between measured and simulated normalized radiances as a function of spatial
pixel, and are applied uniformly across all observation times. We also adopt the newly implemented forward model, additional
fitting parameters, and auxiliary data from the OMI Collection 4 ozone profile algorithm (Bak et al., 2024). As a result, version
3.0 achieves a spectral fitting residual of 0.2% (low SZA/VZA) in ozone profile retrievals, indicating a fourfold improvement
compared to version 2.0. Validation results further confirm the improved performance of the version 3.0 ozone profile product.
Comparisons with regular ozonesonde observations from six East and Southeast Asian stations reveal substantial bias reduction
and improved consistency in both the troposphere and lower stratosphere, effectively smoothing the altitude-dependent
oscillating biases observed in version 2.0. The mean tropospheric ozone column bias is reduced from +36.2% in version 2.0
to −7.7% in version 3.0, with the correlation improved from 0.80 to 0.87. Stratospheric retrievals also show good agreement,
with a mean bias of 4.3% and a correlation coefficient of 0.92. Time series comparisons of tropospheric ozone demonstrate a
better representation of the seasonal cycle in version 3.0, whereas version 2.0 exhibited an artificial increasing trend. Additional
validation using ozonesonde data from the 2024 Asia-AQ campaign supports the improved vertical structure and day-to-day
variability captured by GEMS version 3.0. Furthermore, GEMS total ozone columns derived from version 3.0 profiles show
excellent agreement with Pandora measurements (r = 0.99, mean bias= −3.7 DU), outperforming the GEMS total ozone product.

In this study, we focused on the afternoon measurements at 04:45 UTC (13:45 local time, KST), which correspond to

the overpass time of polar-orbiting satellites in East Asia. In the next version (version 4), we will aim to improve and validate
the ozone profile product for hourly observations. Irradiance calibration will be enhanced by accounting for BTDF effects and
optical degradation in the Level 1C processing, which is expected to provide a more robust foundation for both ozone profile
retrievals and auxiliary input data such as total ozone and cloud information. In turn, the use of soft spectra will be extended
to support hourly, seasonal, and yearly applications, enabling improved temporal consistency in the quality of the GEMS ozone
profile product for both diurnal variation analysis and long-term atmospheric monitoring.
**Acknowledgements**

We thank the GEMS science team and the Environmental Satellite Center (ESC) of the National Institute of Environmental

Research (NIER) for their support in the development of the GEMS ozone profile retrieval algorithm. We also acknowledge
the contributions of the TEMPO, ESA PEGASUS, and ASIA-AQ, WOUDC, SHADOZ teams to algorithm improvements and
product validation.

**Competing interests.** The authors have no competing interests

**Code availability**

The GEMS L2 O$_3$P algorithm is not available publicly.



**Data availability**

GEMS L2 O3P data can be obtained from the Environmental Satellite Center website (https://nesc.nier.go.kr/en/html/datasvc/index.do, NIER, 2025). The Asia-AQ campaign archives are available from https://www-air.larc.nasa.gov/missions/asia-aq/ (NASA, 2023). The regular ozonesonde observations are downloaded from the WOUDC, SHADOZ, and KMA websites.

**Financial support**

This research was supported by Basic Science Research Program through the National Research Foundation of Korea (NRF) funded by the Ministry of Education (grant no. 2020R1A6A1A03044834 and 2021R1A2C1004984). Additional support was provided by a grant from the National Institute of Environment Research (NIER), funded by the Ministry of Environment (MOE) of the Republic of Korea (grant no. NIER-2025-04-02-063). GEMS O3P retrieval characterization and validation studies were performed within the PEGASOS (Product Evaluation of GEMS L2 via Assessment with S5P and Other Sensors) project funded by the European Space Agency (ESA) (contract No. 4000138176/22/I-DT-lr). X.L. and G.G.A were supported by the NASA TEMPO project (Contract No. NNL13AA09C), as well as the NASA Grant 80NSSC19K1626.

**Author Contributions** J.B., D.C., J.K. (Jae-Hwan Kim), X.L., and K.Y. developed the ozone profile retrieval algorithm. G.G.A. developed the radiance data reading modules. A.K. and J.C.L. performed the retrieval characterization. J.H.K. (Ja-Ho Koo) and J.K. (Joowan Kim) provided the Asia-AQ ozonesonde data. S.H., K.B., Y.J and K.P.H. conducted the validation. C.H.K., H.L., and W.J. advised on the implementation of meteorological reanalysis and forecast data. J.K. (Jhoon Kim) led the overall GEMS project. H.H. and W.L. managed the project. All authors contributed to the data analysis and manuscript preparation.

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
