# Peer review of "GEMS ozone profile retrieval: impact and validation of version 2 3.0 improvements"

_EGUsphere, 2025_

## Referee Comment (RC1)

**General remarks**

Juseon Bak et al. provide the first comprehensive characterization of the GEMS operational ozone profile retrieval, including detailed descriptions of the spectral and radiometric calibration enhancements introduced in the version 3.0 algorithm. These improvements are fundamental for the exploitation of the full potential of the geostationary ozone profile data, not only for application in air-quality monitoring but also for investigating local transport processes. Furthermore, the calibration methodologies described and the retrieval show cases are a valuable reference for the ozone profile retrieval community. Therefore, I believe this work aligns closely with the AMT journal's scope and I recommend the publication after addressing the comments below.

**Introduction**

- Is a reference available demonstrating the capabilities of the OMI ozone profile product (cited at lines 62-64)?
- It is not very clear to me the citation at line 67 of (Keppens et al., 2024), which does not explicitly refer to GEMS L2 assessment in the context of the PEGASOS project. Is there an explicit reference available for the PEGASOS project?
- Please look into line 642, which seems to contain an incorrect doi of 2015

**Section 2.4** When I first read the manuscript it was not clear to me that the algorithm updates would be described in this section. Could it be an idea to add "Implementation details and algorithms updates"? It seems to me that this section is mostly about the algorithm updates introduced in v3.0.

**Figure 7** I find the choice of these retrieval show cases very helpful to understand the retrieval capabilities. I find the higher DFS (orange) region in Fig.7b and lower DFS (darker green, around Vietnam-Laos) in Fig.7c quite interesting, but they don't seem to be correlated to their corresponding ozone distribution in Suppl. Fig. 2. However, it seems that the higher DFS region in Fig. 7b has some correlation to Fig. S2 (d), which I was curious to know if you were expecting it.

**Figure 8** I was not expecting this behavior for the retrieval offset, looking at the quite well-behaved shape of the averaging kernel, shown in Figure 6. If I compare this figure with the one of the TROPOMI ozone profile (Fig. 11a, Keppens et al., 2024), the offset trend is driven by the higher information content, but this doesn't seem the case for

Figure 8. So I am wondering if this could be related to the different algorithm settings or if it is more instrument related.

**Conclusions** It might be helpful to add some information regarding the time of the implementation of the updates of version 3 in the operational stream, or if they are already publicly available.

**Minor comments**

- Line 76 I think the "and" can be omitted in point (3)
- Line 297, repetition of Fig. 7c and the first one should refer to Fig.b as the text refers to the troposphere
- Reference Bak et al (2019) in line 347 is missing

---

## Author Comment (AC1)

**Responses to comments from Review # 1**

**■ General Comment**

Juseon Bak et al. provide the first comprehensive characterization of the GEMS operational ozone profile retrieval, including detailed descriptions of the spectral and radiometric calibration enhancements introduced in the version 3.0 algorithm. These improvements are fundamental for the exploitation of the full potential of the geostationary ozone profile data, not only for application in air-quality monitoring but also for investigating local transport processes. Furthermore, the calibration methodologies described and the retrieval show cases are a valuable reference for the ozone profile retrieval community. Therefore, I believe this work aligns closely with the AMT journal's scope and I recommend the publication after addressing the comments below.

**Reply to general comments**: We sincerely thank Dr. Serena Di Pede for the positive and encouraging comments. We are pleased to hear that the reviewer finds our work relevant to the AMT journal's scope and a valuable contribution to the ozone profile retrieval community. We have carefully addressed all the comments provided below.

**■ Specific Comments**

**Comment #1** Is a reference available demonstrating the capabilities of the OMI ozone profile product (cited at lines 62-64)?

➔ **Reply** Thank you for pointing this out. As detailed in Bak et al. (2024), "The OMPROFOZ product has contributed to a better understanding of chemical and dynamical ozone variability associated with anthropogenic pollution over central and eastern China (Hayashida et al., 2015; Wei et al., 2022), transport of anthropogenic pollution in free troposphere (Walker et al., 2010) and stratospheric ozone intrusion (Kuang et al., 2017) as well as ozone concentration changes in the Asian summer monsoon (Lu et al., 2018; Luo et al., 2018). Moreover, this product has been used to quantify the global tropospheric budget of ozone and to evaluate how well current chemistry-climate models reproduce the observations (Hu et al., 2017; Zhang et al., 2010)." To address the reviewer's comment, we have revised the manuscript to cite Bak et al. (2024) and the references therein.

**Comment #2**. It is not very clear to me the citation at line 67 of (Keppens et al., 2024),

which does not explicitly refer to GEMS L2 assessment in the context of the PEGASOS project. Is there an explicit reference available for the PEGASOS project?

➔ **Reply** Thank you for pointing this out. The citation of Keppens et al. (2024) was incorrect in this context. The PEGASOS validation report is an internally shared document and is not publicly available as a formal publication. Therefore, we have replaced the citation with the official project webpage: https://www.dlr.de/en/eoc/research-transfer/projects-missions/pegasos.

**Comment #3** Please look into line 642, which seems to contain an incorrect doi of 2015

➔ **Reply** The reference has been corrected.

**Section 2.4** When I first read the manuscript it was not clear to me that the algorithm updates would be described in this section. Could it be an idea to add "Implementation details and algorithms updates"? It seems to me that this section is mostly about the algorithm updates introduced in v3.0.

➔ **Reply** Thank you for your insightful suggestion. To improve clarity and guide the reader more effectively, we have revised the section title to "Implementation Details and Algorithm Updates."

**Figure 7** I find the choice of these retrievals show cases very helpful to understand the retrieval capabilities. I find the higher DFS (orange) region in Fig.7b and lower DFS (darker green, around Vietnam-Laos) in Fig.7c quite interesting, but they don't seem to be correlated to their corresponding ozone distribution in Suppl. Fig. 2. However, it seems that the higher DFS region in Fig. 7b has some correlation to Fig. S2 (d), which I was curious to know if you were expecting it.

➔ **Reply** Thank you for your thoughtful comment and for giving us the opportunity to enhance the analysis. In the stratosphere, the DFSs enhance with longer light path length. The darker green (Fig 7c), as noted by the review, corresponds to areas with small solar zenith angles (SZA < 10°), which is also evident in Fig. 7d. In the troposphere, while DFS is also influenced by the light path length, the relationship becomes more complex due to additional factors such as tropospheric ozone amount, surface reflectance, cloud optical properties, aerosol scattering, and other scene-dependent characteristics. As shown, tropospheric DFSs show a significant correlation with tropospheric ozone amounts in regions where SZA is not extreme. On December 16, 2021, the high DFS values (~0.8) appear to be associated with elevated surface albedo. This high albedo was retrieved

because the cloud input was missing in the ozone profile retrieval. As shown in the following figures, the GEMS cloud l2 product did not provide valid output over the central region of the large-scale cloud system identified from RGB image. In the absence of cloud information, the retrieval algorithm estimated a higher surface albedo to fit the observed radiance. We have added the relevant figures to the Supplement (Figure S3) and revised the manuscript to include this explanation.

[Figure]

**Figure 1. (left) effective cloud fraction taken from GEMS L2 cloud product on 16 December 2021; (center) surface albedo fitted from GEMS L2 ozone profile product; (right) RGB image from the VIRR instrument.**

**Figure 8** I was not expecting this behavior for the retrieval offset, looking at the quite well-behaved shape of the averaging kernel, shown in Figure 6. If I compare this figure with the one of the TROPOMI ozone profile (Fig. 11a, Keppens et al., 2024), the offset trend is driven by the higher information content, but this doesn't seem the case for Figure 8. So I am wondering if this could be related to the different algorithm settings or if it is more instrument related.

➔ **Reply.** The retrieval offset represents a mismatch between the peak altitude of averaging kernel and the altitude the retrieval is intended to capture. Ideally, the peak of each averaging kernel should align with the corresponding retrieval grid, assuming sufficient vertical sensitivity. However, GEMS uses a more limited spectral range (310–330 nm) compared to TROPOMI (270–330 nm). As shown in Fig. 11a of Keppens et al. (2024), TROPOMI retrievals exhibit nearly zero offset between 20 km and 50 km, benefiting from the Hartley ozone band (270–310 nm), which provides strong height-resolved information through Rayleigh scattering. In contrast, the GEMS averaging kernels generally peak between 15 and 25 km, and above this range, the retrieval offset increases negatively due to the lack of sufficient information at higher altitudes. In the troposphere, both GEMS and TROPOMI exhibit similar offset behavior, with increasing values toward the surface.

**Conclusions** It might be helpful to add some information regarding the time of the

implementation of the updates of version 3 in the operational stream, or if they are already publicly available.

➔ We completely agree — this is an important point that should have been mentioned. The reprocessing has been completed for the entire mission period, and the GEMS v3.0 ozone profile product is now publicly available from November 2020 onward. "*The reprocessing of the GEMS ozone profile dataset has been completed and the version 3 product is publicly available through the Environmental Satellite Center website (https://nesc.nier.go.kr/en/html/datasvc/index.do; NIER, 2025).*" has been added in the conclusion section of this paper.

**■ Minor comments**

**1.Line 76 I think the "and" can be omitted in point (3)**

➔Accepted.

**2.Line 297, repetition of Fig. 7c and the first one should refer to Fig.b as the text refers to the troposphere**

➔ Thank you for your careful review. The issue has been corrected — the first reference now correctly points to Fig. 7b, as it pertains to the troposphere.

**3.Reference Bak et al (2019) in line 347 is missing**

➔*Bak, J., Baek, K.-H., Kim, J.-H., Liu, X., Kim, J., and Chance, K.: Cross-evaluation of GEMS tropospheric ozone retrieval performance using OMI data and the use of an ozonesonde dataset over East Asia for validation, Atmos. Meas. Tech., 12, 5201–5215, https://doi.org/10.5194/amt-12-5201-2019, 2019.* Is added in the reference list.

---

## Author Comment (AC2)

**Responses to comments from Review # 2**

**■ General Comment**

I thank the authors for a well-organized and written discussion of their new ozone profile product. They have clearly laid out the changes from the previous version and how these have led to an improved product. I would, however, like to know a bit more about the calibration choices that were made. I will appreciate if the authors can address my questions below, and where appropriate modify the manuscript text to clarify points regarding their approach.

**Reply to general comments**: Thank you very much for your thoughtful and encouraging comments on our manuscript. We also appreciate your interest in the calibration choices. We will carefully address each of your questions in detail.

**■ Specific Comments**

**Comment1** Section 2.5.2 It's clear that using a measured irradiance that includes seasonal and long-term errors will result in significant ozone errors. What is not clear is the best approach to manage this problem. The authors choice to use daily solar irradiance measurements more or less forces them to find a correction for Working diffuser irradiance measurement errors. Have the authors performed a trade study that indicates this is really the optimum solution? An often-heard claim is that normalization using daily solar measurements is required in order to adequately account for detector variations and anomalies. Such an assertion must really be demonstrated for each instrument, and actually for each product. What would the ozone product performance be if the authors utilized a GEMS irradiance fixed in time near the start of the mission? Does all the extra effort creating daily corrections really improve the product compared with simply normalizing by a Day 1 solar?

➔ **Reply1.1** We appreciate the reviewer's insightful comment, and we believe the reviewer has deep expertise in irradiance calibration issues. We believe, adopting a single Day 1 reference would require assurance that radiance degradation does not occur over time and that no unintended cancellation effects with the common degradation behavior of the optical elements in radiance and irradiance measurements. Most importantly, such an approach would only be feasible if the BTDF-induced geometry effects were negligible. In our view, the use of a fixed irradiance could be a viable option once the BTDF angular-dependent issue is fully resolved and temporal soft calibration is reliably applied. The L1B team is currently preparing the development of a BTDF correction, will help make such an approach practically achievable. Based on this comment, we plan to first test the fixed irradiance approach with long-term OMI ozone profile retrievals with dense ozonesonde measurements (e.g., Uccle, with two to three launches per week). Depending on the outcome, we will then consider applying this approach to GEMS in preparation for the next version.

➔ **Reply1.2**. To optimize ozone profile retrievals, we extensively performed a trade study to evaluate different radiometric correction methods. In version 3, we ultimately adopted the use of a scaling factor for irradiance together with a soft calibration spectrum

(dependent only on the CCD dimension) applied to normalized radiance. In earlier tests, we also experimented with polynomial fitting to reduce offsets in irradiance:

$$F = F_G - \sum_{0}^{N-1} P_b(i)(\lambda - \bar{\lambda})^{i-1} \; (N = 0 \dots 3)$$

As a result, the first-order correction was considered most effective (Fig1.). However, its performance was not consistent across seasons. In particular, the middle-stratospheric ozone is severely underestimated across seasons (Fig.2). In preparation for version 3.0, we also tested the multiple irradiance beta data provided by the GEMS L1C team, but its application further degraded our retrievals. Therefore, we ultimately decided to apply a combined approach of scaling and soft calibration. In the manuscript, the polynomial experiment results were omitted to maintain a clear focus on the GEMS v3.0 operational version. It briefly mentions "*This decision was made because applying the baseline polynomial Pb directly to the irradiance introduced artificial structures into the spectral fitting of the normalized radiance, resulting in a significant underestimation of stratospheric ozone retrievals.*"

[Figure]

Fig 1. Tropospheric ozone retrievals using different polynomial orders for irradiance correction (note that soft calibration was not applied). Daytime is 20220615_0445.

[Figure]

Fig.2. Ozonesonde observations during the Asia-AQ 2024 campaign (Feb/blue–Mar/red) and the corresponding GEMS version 2.0 and version 3.0 retrievals. A first-order offset correction is applied as the traditional approach.

**Comment2** Regardless of which method is used to generate the solar irradiance, there is no discussion of how long-term radiometric changes in the instrument are accounted for. This is not something soft calibration is capable of dealing with. The authors fail to discuss which of the two GEMS solar measurements they are using, Reference diffuser or Working diffuser. But given the temporal density of Fig. 3, I can assume they are using Working. Why not use the Reference instead and avoid most of the diffuser degradation? BTDF issues can be dealt with by choosing Reference measurements at similar solar incidence angles. This will typically yield two useful solar measurements per year, which can then be used to interpolate in time. I can understand that this approach may not work as well for the ozone product as the method the authors have chosen. But have the authors considered such alternatives? The authors should discuss alternative approaches and why they believe the chosen approach works

best. They should also the discuss the drawbacks (i.e. long-term trends) with their chosen approach.

**Rely3**

- Thank you for this comment. We had no option to choose between the reference and working irradiance, as the reference diffuser data was not shared with the L2 team. Instead, the L1C team is preparing an update to the irradiance calibration that will address both geometry dependence and degradation
- However, applying a scaling correction to the daily irradiance measurements is expected to help mitigate the degradation issue. Figure 3 (manuscript) shows that the correction value dynamically varies with time and geometry. Demonstrating the improvement between version 2 and version 3 with respect to degradation is challenging due to the limited validation data from ozonesondes. Nevertheless, our companion paper (Hong et al., under review) demonstrates that GEMS version 3.0 ozone profiles, when integrated to total ozone, show reduced impact compared with GEMS version 2.1 total ozone, for which no calibration was applied, as validated against Pandora data.

[Figure]

Fig.3 Comparison between satellite total ozone products and pandora observations (Hong et al. under review). O3P and O3T indicates GEMS v3.0 ozone profile and GEMS v2.1 total ozone products.

**Comment3** Section 3, Line 334 I had to read this line several times before I understood that the authors are referring to a specific viewing condition when they use the phrase "sideways solar irradiance". I recommend using more precise terminology (e.g. involving SolZA) so as not to confuse the readers.

Rely3. According to this comment, we have revised Line 334 as "The offset decrease under more oblique solar and viewing geometry, although this is accompanied by a reduction in tropospheric retrieval sensitivity."

The offset is reduced for more sideways solar irradiance and observation of the troposphere, although it has to be taken into account that the tropospheric retrieval sensitivity is at the same time reduced as well (see above).

**Comment 4** Figure 13. This figure is somewhat confusing. It is not possible to see any data points from OMPS, yet there are some faint points with an unknown color (in the 330-390 DU range) that are not assigned to any mission. Please provide a better figure or clear up the confusion by describing in the figure caption.

.

- We have revised Figure 13 to clearly display the OMPS data points. During February and March 2024, there were unprecedentedly frequent rain events, and the available Pandora data cover 44 days. After quality assurance, 42 valid days were retained. As shown in Figure below, no data points fall within the 370–380 DU range.

[Figure]

**Figure 3. Pandora observations in Feb-Mar 2024.**

- In caption of Fig .13, "*A total of 44 Pandora observation days was available, of which 42 remained after quality control.*" has been added.

---

## Author Response (AR2)

**Responses to comments from Review # 2**

In second round

**Comment from Editor**: Dear Juseon et al., R#2 has some comments on your revised paper that require attention. I think it is important that choices made in the calibration are properly evaluated, among others the limitations clearly addressed and discussed. Best wishes, Mark

**Reply to Editor**: We appreciate the valuable feedback from both you and Reviewer #2, and we will carefully revise the manuscript to more thoroughly address the limitations as suggested

**Specific comment #1:**

I understand that the authors' approach to soft calibration of the GEMS Level 1 product is constrained by the contents of that product. I recommend that they include more information about *the state of the Level 1C product* they are using so that the readers can better understand their chosen soft calibration approach.

**Reply to comment #1:**

As mentioned in Section 2.1 (GEMS operations), "*Currently, Version 2 irradiance and Version 1.2.4 radiance products are used as the standard Level 1C inputs for subsequent Level 2 processing. Neither product has been reprocessed since the initial on-orbit testing, and the official data period began on November 1, 2020.*"

Kang et al. (2024) reported on the status of the GEMS Version 2 irradiance, highlighting geometry-dependent biases resulting from the missing BTDF correction and the effects of time-dependent degradation. The newly implemented calibration process in the GEMS ozone profile Version 3.0 provides a clearer representation of the irradiance status in the 310–330 nm range as following:

**In 2.5.1 Spectral correction**, as shown in Figure 2, substantial discrepancies are evident in both the magnitude and spatial pattern of the spectral shift between radiance and irradiance, ranging from 0.02 to 0.04 nm, with larger differences toward the northern edge of the spatial domain. Additionally, as degradation progresses, pixel-to-pixel perturbations

increase toward the central spatial pixels in both radiance and irradiance measurements. Therefore, independent shift correction is implemented to radiance and irradiance. To ensure computational efficiency, the radiance shift is determined from the first mirror step and applied uniformly along the scan direction, based on the observation that spectral shifts in the radiance data remain relatively uniform across mirror steps.

In **2.5.2 Radiometric correction,** the GEMS irradiance is **spatially and seasonally biased** due to a missing calibration component for the BTDF, which defines how light transmits through a diffuser based on incident and outgoing angles—a well-known issue (Kang et al. 2024; Bak et al. 2025b). Additionally, Bak et al. (2025b) identified **progressive radiometric degradation**, resulting in an annual irradiance decrease of **~5%** in the shorter UV range. They also reported that the measured irradiance is roughly **40%** lower than the solar reference near 325 nm. ➔ status of radiometric accuracy

As presented **in Figure 3**, the derived values of $C$ exhibit significant seasonal and spatial variations in irradiance offset related to angular dependence, along with a gradual temporal decline attributable to optical degradation, most prominently at the middle spatial pixels. In version 3, only the scaling factor $C$ is applied in the irradiance correction, by dividing the irradiance by C.

Residual wavelength-dependent uncertainties are instead addressed through the soft calibration process, which has been newly implemented in version 3.

**Specific comment #2:**

Firstly, it seems that the Level 1 irradiance product derives only from the Working diffuser and has no calibration correction to account for time-dependent degradation of the instrument or diffuser. Secondly, the Earth radiances reported also have no time-dependent corrections applied. These facts leave the Level 2 products with few good options to deal with instrumental changes. One option involves use of the Reference diffuser data, but apparently that option is not open to the Level 1C user.

**Reply to comment #2:**

Yes, as we addressed in the previous revision, we had no option to choose between the reference and working irradiance, as the reference diffuser data was not shared with the L2 team. Instead, the L1C team is preparing an update to the irradiance calibration that will address both geometry dependence and degradation. And I would like to mention that this work deal with the operational product. Therefore, we should use the common radiance and irradiance inputs that are applied consistently in all L2 processing.

**Specific comment #3:**

The authors have chosen one approach to soft calibration that has its own unique set of problems. I encourage the authors to identify and acknowledge those problems in their paper. First and foremost is that they are effectively normalizing the GEMS solar irradiance measurements to solar irradiance reference standard. Therefore, any construction of a BSDF measurement quantity will involve a denominator that does not include instrument drift and a numerator that does include instrument drift. The resulting BSDF values will drift in time as the instrument response degrades. It will be helpful if the authors can acknowledge this and estimate the magnitude of this error in the paper.

**Reply to comment #3:**

We appreciate the reviewer's careful consideration of the limitation inherent to our soft-calibration approach. We agree that caution is needed when applying the empirical method, we have used, which involves scaling corrections based on the normalization of irradiance to a reference and soft calibration applied to the normalized radiance. As review mentioned, the irradiance correction might lead to over-correction, thereby preventing the cancellation of radiometric errors that are present in both the radiance and irradiance. Therefore, we apply a simple correction scaling value corresponding the sum(irradaiance)/sum(reference). In earlier tests, we also experimented with polynomial fitting to reduce wavelength dependent offsets in irradiance:

$$F = F_G - \sum_0^{N-1} P_b(i)\big(\lambda - \bar{\lambda}\big)^{i-1} \ (N = 0 \dots 3)$$

We found that higher-order corrections tend to over-correct the irradiance. Therefore, we applied a single scaling value to reduce the irradiance offset, followed by soft calibration to mitigate the wavelength-dependent biases in the normalized radiance. We acknowledge that our approach does not perfectly address degradation and other calibration issues; however, it represents a clear improvement compared to previous versions. Evaluating the long-term consistency of ozone profiles is particularly challenging for GEMS due to the limited availability of ozonesonde data. Nevertheless, our companion paper (Hong et al., under review) demonstrates the reliability of integrated column ozone from the ozone profiles through comparisons with Pandora measurements.